# Position: Sustainable Open-Source AI Requires Tracking the Cumulative Footprint of Derivatives

**Shaina Raza**[1]  **Iuliia Zarubiieva**[1 2]  **Ahmed Y. Radwan**[1]  **Nathaniel Lesperance**[1 2]  **Deval Pandya**[1]
**Sedef Akinli Kocak**[1]  **Graham W. Taylor**[1 2]

## Abstract

Open-source AI is scaling rapidly, and model hubs now host millions of artifacts. Each foundation model can spawn large numbers of fine-tunes, adapters, quantizations, merges, and forks. **We take the position that compute efficiency alone is insufficient for sustainability in open-source AI. Lower per-run costs can accelerate experimentation and deployment, increasing aggregate footprint unless impacts are measurable and comparable across derivative lineages.** However, the energy use, water consumption, and emissions of these derivative lineages are rarely measured or disclosed in a consistent, comparable way, leaving aggregate ecosystem impact largely invisible. We argue that sustainable open-source AI requires a coordination infrastructure that tracks impacts across model lineages, not only base models. We propose **Data and Impact Accounting (DIA)**, a lightweight, non-restrictive transparency layer that (i) standardizes carbon-and-water reporting metadata, (ii) integrates low-friction measurement into common training and inference pipelines, and (iii) aggregates reports via public dashboards to summarize cumulative impacts across releases and derivatives. DIA makes derivative costs visible and supports ecosystem-level accountability while preserving openness. 🌐 Project page

## 1. Introduction

The open-source artificial intelligence (AI) ecosystem has grown rapidly. Hugging Face hosts over 2 million models, datasets, and applications (Hugging Face, 2024). A single foundation model like Meta Llama can spawn hundreds of publicly documented derivatives within months of release (Oderinwale et al., 2025). This has helped democratize AI, allowing researchers to adapt models without having the resources to train from scratch. This success, however, creates a coordination problem that is invisible at the individual level. Every derivative model costs energy and compute to produce. For example, parameter-efficient methods such as Low-Rank Adaptation (LoRA) (Hu et al., 2022) and Quantized Low-Rank Adaptation (QLoRA) (Dettmers et al., 2023) fine-tune large models by updating only a small set of additional parameters rather than retraining the full network. Individually, the cost of a single fine-tune appear modest compared to pretraining. However, collectively, across thousands of downstream derivatives, these cumulative costs can exceed the base model investment.

This mirrors **the tragedy of the commons** (Hardin, 1968), where individual actions like fine-tuning can collectively increase energy and water use. Here, the commons are the atmosphere and freshwater resources. Carbon emissions and water consumption from AI training and deployment impose costs that are external to any single actor but accumulate across the ecosystem and degrade shared resources. The open-source ecosystem currently lacks governance mechanisms to coordinate responsible resource use. Figure 1 highlights that AI's footprint extends beyond base training to include downstream derivatives.

Per-model efficiency gains remain important, but without ecosystem-level coordination, these savings can be offset by increased use. Methods such as distillation (Wang et al., 2022), pruning (Tmamna et al., 2024), mixed-precision (Dörrich et al., 2023), and data subset selection (Killamsetty et al., 2021) reduce the cost of training and inference, but lower costs often lead to more experimentation and deployment, increasing total emissions. Economics and energy studies describe this as a **rebound effect** (Özsoy, 2024), where efficiency improvements encourage greater usage, and in extreme cases as the **Jevons Paradox** (Sharma, 2024), where overall emissions rise despite per-model gains.

Rebound effects are a well-established area of study. The

---

[1]Vector Institute for Artificial Intelligence, MaRS Centre, Toronto, ON M5G 1L7, Canada [2]University of Guelph, School of Engineering, Guelph, ON N1G 2W1, Canada. Correspondence to: Shaina Raza <shaina.raza@vectorinstitute.ai>.

*Proceedings of the 43rd International Conference on Machine Learning*, Seoul, South Korea. PMLR 306, 2026. Copyright 2026 by the author(s).

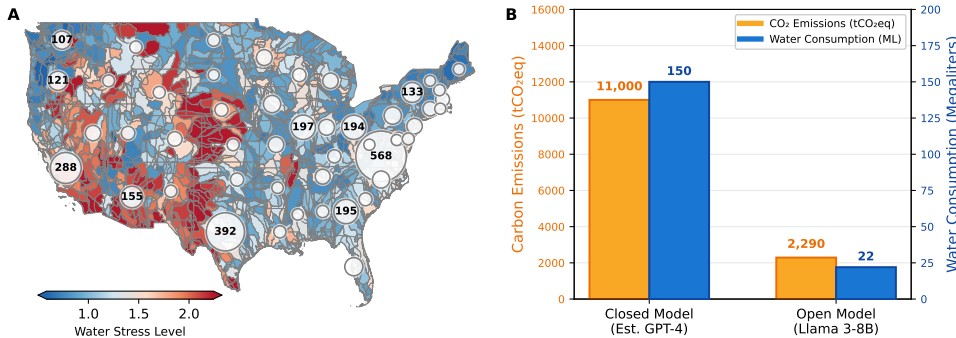

*Figure 1.* The hidden environmental reality of the AI ecosystem. **(A)** Localized water stress across the United States, according to World Resource Institute (2023). Circles illustrate the number of data centres per state (https://www.datacentermap.com/usa/), with the top 10 states labelled with their respective counts. Texas (392), California (288), and Arizona (155) are among the states that have both a high number of data centres and high water stress levels. **(B)** Estimated order-of-magnitude comparison of training-related carbon and water footprints. Closed-model values (e.g., GPT-4) are approximate and based on secondary public estimates rather than audited disclosures (IEA, 2025). Open-model values (e.g., Llama 3) are drawn from official Meta documentation (Dubey et al., 2024) when available. Water consumption values for both model types are estimated using reported/inferred energy consumption and average water usage effectiveness (WUE) factors.

literature distinguishes direct, indirect, and economy-wide rebounds, capturing behavioural and productivity-driven responses that offset expected savings (Özsoy, 2024). Research in software engineering and Information and Communications Technology (ICT) sustainability shows that design, configuration, and architectural choices directly influence hardware energy consumption (Becker et al., 2015). Green software engineering studies confirm these practices measurably affect energy footprints (Procaccianti et al., 2016).

Given these dynamics, our position focuses on open-source ecosystems, where the tragedy of the commons arises most clearly due to the absence of coordination and visibility. **We take the position that sustainable open-source AI requires ecosystem-level impact accounting across model lineages and derivatives, not only per-model efficiency improvements.** Closed-source model developers operate within organizations that may be subject to general corporate sustainability reporting requirements (European Parliament and Council of the European Union, 2022; California State Legislature, 2023a;b) and publish voluntary environmental disclosures (Google, 2025; Microsoft, 2024; Meta, 2025). However, model-level training emissions are not currently mandated under these frameworks. Open-source ecosystems, by contrast, lack even these partial accountability mechanisms, making collective action problems more acute and coordination infrastructure essential.

## 2. Empirical Background: The Rebound Effect in AI

### 2.1. Efficiency Gains Are Real But Insufficient

Modern optimization techniques can substantially reduce per-run compute and energy costs. For example, 8-bit quan-

tization can reduce model memory by ∼4× and speed up inference in practice (Krishnamoorthi, 2018). Knowledge distillation can produce smaller models that retain most of the original performance; for instance, DistilBERT preserves about 97% of BERT capabilities while being faster at inference (Sanh et al., 2020). QLoRA further enables fine-tuning of 65B-parameter models on a single 48GB GPU (Dettmers et al., 2023). However, real-world inference energy varies widely across model sizes and serving stacks, with benchmarking showing large differences across inference engines and settings (Niu et al., 2025; Desislavov et al., 2023). Moreover, footprint reporting is often inconsistent across studies (Henderson et al., 2020), and the lack of standardized assumptions makes ecosystem-level comparisons difficult (Patterson et al., 2021).

Despite much LLM optimization, aggregate consumption continues to rise. The International Energy Agency (IEA) projects that global data centre electricity consumption will double from approximately 415 TWh in 2024 to 945 TWh by 2030, a 15% annual growth rate, four times faster than total electricity demand (IEA, 2025). AI-specific accelerated servers are growing at 30% annually. In the United States alone, AI servers consumed 53-76 TWh in 2024, projected to reach 165-326 TWh by 2028 (Carbon Capture, 2024; O'Donnell & Crownhart, 2025). This pattern is consistent with a rebound effect, where efficiency improvements reduce the cost of AI inference, which increases demand and, in turn, drives more supply that ultimately overwhelms the efficiency gains and raises total energy consumption.

### 2.2. The Rebound Effect in Open vs. Closed Ecosystems

**Rebound mechanisms in AI:** Efficiency improvements lower energy and cost per query but do not guarantee reduced aggregate impact. In ICT systems, rebound effects

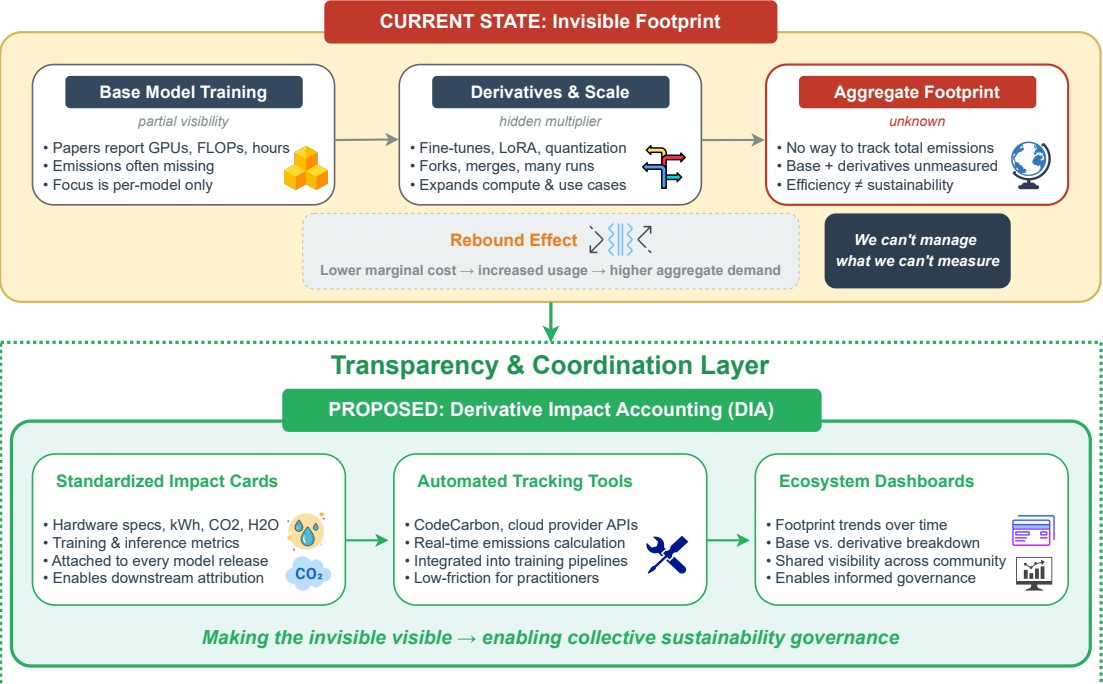

*Figure 2.* Overview of Data and Impact Accounting (DIA). **Top:** Base-model training emissions may be reported, but derivative artifacts (e.g., fine-tunes, LoRA adapters, quantizations, merges) are typically untracked, making aggregate ecosystem impact unobservable. **Bottom:** DIA introduces a low-friction visibility layer with (1) standardized impact reporting in model metadata, (2) automated tracking via existing tools, and (3) ecosystem-level aggregation through public dashboards.

can arise through direct (more use of the same service), indirect (new uses enabled by lower costs), and economy-wide channels (Charfeddine et al., 2024). Similar dynamics have been discussed for cloud computing and AI workloads, where lower marginal compute costs expand experimentation and deployment (Sharma, 2024).

While causal effects cannot yet be firmly established, enabling conditions are in place, for example, per-query energy has fallen to sub-watt-hour levels (Vahdat & Dean, 2025), usage has scaled rapidly (e.g., 18B messages/week by July 2025) (Chatterji et al., 2025), and data-centre electricity demand is projected to grow substantially with AI as a key driver (International Energy Agency, 2024). Recent work frames rebound as a central risk, arguing efficiency alone cannot ensure net reductions without governance and demand-side constraints (Luccioni et al., 2025). Importantly, rebound pathways may differ substantially between open and closed ecosystems.

**Closed model dynamics:** A model like GPT-4 is trained and served centrally via an API. Though exact training electricity use is not disclosed, third-party analyses place training in the tens of GWh (Chen et al., 2025; IEA, 2025). Inference then scales with demand, but remains centrally metered in provider data centres. For example, OpenAI reports over 2.5B ChatGPT messages per day (OpenAI, 2025); com-

bined with per-query energy estimates, inference represents a substantial ongoing footprint (You, 2025).

**Open model dynamics:** Once released, an open model like Meta Llama 3 branches into many derivatives produced by independent users, including fine-tunes, quantizations, adapters, merges, and distilled variants. This diffuses environmental impacts across a distributed ecosystem, making the aggregate footprint harder to quantify (Oderinwale et al., 2025). Meta reports that pretraining Llama 3 (8B and 70B combined) emitted approximately 2,290 $tCO_2eq$ (Dubey et al., 2024). Derivative proliferation is already visible at scale; for example, Oderinwale et al. (2025) documents 146 derivatives for a single model family. Even if most derivatives are cheap, the aggregate emissions across hundreds can exceed base model training by multiples. Precise estimation is impossible because derivative compute is rarely disclosed. This motivates our position for a coordination mechanism as proposed in Section 4 (Figure 2).

### 2.3. Estimating the Hidden Footprint

Table 1 summarizes training emissions for major models (2020-2024). Following established ML carbon accounting methodology (Strubell et al., 2019; Lacoste et al., 2019; Patterson et al., 2021), we estimate electricity use, carbon emissions, and water consumption when direct reporting

*Table 1.* Training emissions and water consumption of selected GenAI models (2020-2024). Models marked with $^\star$ have publicly released weights. Tree equivalent assumes 25 kg $CO_2$/tree/year. Water in megalitres (ML; $1\,\text{ML} = 10^6$ L). R = disclosed by the model developer/project report; Est. = estimated by us from disclosed compute/energy; N/D = not disclosed. See Appendix B for detailed source notes, estimation assumptions, and formulae.

| Model[a] | Year | Params | tCO₂eq[b] | R/Est. | Tree Eq. | Water (ML)[c] |
|---|---|---|---|---|---|---|
| GPT-3 | 2020 | 175B | 552 | Est. | 22,080 | 2.5–5.6 |
| BLOOM$^\star$ | 2022 | 176B | 24.7–50.5 | R | 988–2,020 | 0.78–1.73 |
| OPT$^\star$ | 2022 | 175B | 75 | Est. | 3,000 | 0.6–1.3 |
| Falcon 180B$^\star$ | 2023 | 180B | $\sim$1,200[d] | Est. | $\sim$48,000 | 5.0–11.0 |
| Llama 2$^\star$ | 2023 | 70B | 539 | R | 21,560 | 2.4–5.3 |
| Mistral 7B$^\star$ | 2023 | 7.3B | N/D | N/D | N/D | N/D |
| GPT-4 | 2023 | N/D | 4,240–18,870[e] | Est. | 169,600–754,800 | 76–170 |
| Llama 3$^\star$ | 2024 | 8B / 70B | 2,290 | R | 91,600 | 10.2–22.6 |
| Llama 3.1$^\star$ | 2024 | 405B | 8,930 | R | 357,200 | 40–88 |
| DeepSeek-V3$^\star$ | 2024 | 671B | $\sim$545[g] | Est. | $\sim$21,800 | 1.9–4.3 |

is incomplete. When only GPU time is disclosed, we approximate energy as $E = H_{\text{GPU}} \cdot P_{\text{avg}} \cdot \text{PUE}/1000$, where $H_{\text{GPU}}$ is total GPU-hours, $P_{\text{avg}}$ is average GPU power draw (W), and PUE is power usage effectiveness (Masanet et al., 2020). If measured power is unavailable, we use vendor *thermal design power* (TDP) as an upper bound. TDP is the vendor-specified power envelope, and actual draw may be lower depending on utilization; we assume 60-80% of TDP (Henderson et al., 2020; Dodge et al., 2022).

Carbon emissions are computed as $C = E \cdot CI/1000$, where $CI$ is grid carbon intensity (kgCO₂/kWh). We report water consumption using combined water-usage effectiveness (WUE) (Li et al., 2025) and include tree-equivalent values as a rough interpretability aid (U.S. EPA, 2024; Nowak et al., 2013). Full formulae appear in Appendix A.

### 2.4. The Water Dimension: A Hidden Cost

Beyond carbon emissions, AI also consumes substantial amounts of water. We highlight water use as a second environmental externality that is highly localized in Figure 1(A), yet often remains an overlooked and underreported cost of AI infrastructure. Data centres use water for cooling and indirectly for electricity generation. Many facilities use evaporative cooling, where water is consumed (lost to the atmosphere) to dissipate heat. It is important to distinguish water *withdrawal* (water taken from a source) from *consumption* (water not returned locally), since the latter drives local scarcity impacts. A mid-sized data centre can use approximately 1.1 megalitres per day ($\sim$ 300,000 gallons)of water (roughly comparable to $\sim$1,000 households), while hyperscale facilities can consume approximately 19 megalitres per day ($\sim$5 million gallons) (Kane, 2025).

Unlike carbon, water impacts are local and depend on basin-level scarcity. Many data centres are in water-stressed regions, competing with agricultural and residential use. MSCI (Morgan Stanley Capital International) analysis of

14,000 data center assets found one in four may face increased water scarcity by 2050 (MSCI Research and Insights, 2025). Google's Council Bluffs, Iowa data center consumed approximately 4,900 megalitres ($\sim$1.3 billion gallons) of potable water in 2024, making it one of Google's most water-intensive sites. (NASUCA & Schneider Electric, 2025) We model water use with total water usage effectiveness (WUEtotal, L/kWh), capturing on-site cooling and upstream water consumption from electricity generation. (Azevedo & The Green Grid, 2011) Here, WUEtotal refers to consumption (water not returned to the source), not withdrawal, since consumptive use is more directly linked to local scarcity impacts. (Mytton, 2021).

## 3. Position Statement

**Compute efficiency is necessary but not sufficient to reduce AI's *aggregate* environmental footprint. The missing ingredient is ecosystem-level coordination: open-source AI needs a lightweight, standardized carbon-and-water accounting infrastructure to make environmental impact measurable, comparable, and actionable.** Beyond its role in coordination, environmental reporting is also a matter of scientific best practice. Just as the community increasingly expects disclosure of compute budgets, hardware, and training details for reproducibility, reporting the energy and water costs of a contribution is part of responsible and transparent research. Open models enable reproducibility, democratize access, and accelerate research on compression and efficiency techniques that benefit the entire field. Our position is not that open source is the root problem, but rather that *open ecosystems amplify a coordination gap*. When development is distributed across thousands of independent actors, local optimization (cheaper training, faster iterations) can increase total activity even as per-run efficiency improves.

We therefore advocate a minimal but high-leverage inter-

vention: **ecosystem-level footprint visibility**. We propose **Data and Impact Accounting** (**DIA**; Section 4) as a lightweight coordination layer that standardizes how training and deployment costs, such as energy, carbon, and water, are estimated, reported, and aggregated across model families and downstream derivatives. DIA is non-restrictive: it helps open-source communities track cumulative impact, compare alternatives, and avoid a tragedy-of-the-commons outcome in which total footprint grows despite per-model efficiency gains.

Our claim is deliberately narrow. A coordinated sustainability framework in the open-source AI community is necessary, but not sufficient, for addressing climate impacts. We claim only that: (1) current uncoordinated scaling is environmentally fragile, (2) efficiency improvements cannot guarantee aggregate reductions without coordination, and (3) standardized accounting is a necessary first step for responsible, open, and sustainable AI development.

## 4. Proposal: Data and Impact Accounting

We propose Data and Impact Accounting, a lightweight coordination layer for ecosystem-level carbon and water visibility (Figure 2). Next, we explain DIA.

### 4.1. Core components

**(1) A lightweight reporting schema.** DIA defines a minimal footprint schema that can be embedded in a model card or repository metadata. At minimum, it records: (i) hardware type and device count, (ii) training duration (GPU-hours and, optionally, CPU-hours for preprocessing-heavy workflows), (iii) and the estimation method used (e.g., direct measurement via CodeCarbon, hardware-based calculation, or cloud provider API), (iv) estimated water use (L) or facility WUE (L/kWh), (v) grid carbon intensity (kgCO$_2$/kWh) or training region as a proxy, and (vi) model lineage (base model(s) and major downstream derivatives, when applicable). For inference, direct tracking is more challenging because deployment is decentralized across diverse hardware and configurations. DIA addresses this through complementary mechanisms: (vii) standardized per-query energy benchmarks measured under reference conditions (e.g., tokens-per-joule on specified hardware) to estimate local footprint; (viii) optional aggregate usage reporting by inference providers and model hub operators (e.g., download counts, API call volumes); and (ix) deployment efficiency metadata that allows downstream users to project costs under their specific configurations.

**(2) Low-friction instrumentation.** DIA reduces reporting burden by integrating automated measurement tools (e.g., CodeCarbon (Courty et al., 2024), ML CO$_2$ Impact Calculator (Lacoste et al., 2019), and cloud-provider sustainability

APIs) to generate reports with minimal manual effort. For inference benchmarking, standard protocols such as MLPerf Inference (Reddi et al., 2020) can be incorporated into the reporting pipeline to ensure comparable measurements across deployments. For water reporting specifically, we acknowledge that facility-level WUE data is often unavailable to end users. DIA allows region-based defaults with an explicit data quality tier and uncertainty range (or a clearly flagged "data unavailable" designation), preserving comparability without imposing audit-grade requirements.

**(3) Ecosystem-level aggregation.** DIA supports aggregation via a public registry or dashboard that summarizes reported footprints across releases. This enables trend analysis, identification of high-impact model families, and benchmarking of efficiency improvements over time. For inference, aggregation of download statistics and voluntary provider reporting enables estimation of deployment-phase impacts at the ecosystem level, even when per-query tracking is infeasible. Existing model hubs (e.g., Hugging Face) are natural candidates to host these summaries.

### 4.2. Design principles

DIA is voluntary and low-friction, with adoption driven by social incentives and community norms, similar to how model cards and dataset documentation became common (Mitchell et al., 2019; OECD.AI, 2023). DIA accepts that early measurements will be imperfect. Approximate estimates based on hardware and duration are sufficient for directional insight because the goal is visibility into trends and relative impacts rather than auditing individual projects. Importantly, DIA preserves open-source benefits by avoiding barriers to entry; small teams can provide minimal information, and the framework focuses on aggregate patterns rather than policing individual contributions. Finally, by making efficiency visible and comparable, DIA creates a positive feedback loop.

### 4.3. Implementation path

We envision a phased rollout. **Phase 1: Norm-setting.** Major open-source labs adopt standardized reporting for flagship releases, and conferences encourage emissions reporting in submissions (e.g., through reproducibility, transparency or ethics checklists). **Phase 2: Friction reduction.** Common training stacks (e.g., PyTorch, JAX, TensorFlow and Transformers) expose optional emissions tracking by default, and cloud providers surface location-adjusted carbon and water information in job summaries. **Phase 3: Ecosystem visibility.** Model hubs and community dashboards aggregate and display reported data, enabling researchers and practitioners to query footprint estimates for model families and track ecosystem trends over time. **Phase 4: Accountability.** DIA becomes part of routine ecosystem work-

flows via non-binding badges or "impact labels" on model pages, standardized citations for impact statements, and benchmarking that supports voluntary targets and progress tracking.

Consider the Llama 3 ecosystem: 146 documented derivatives (Oderinwale et al., 2025), with per-derivative costs ranging from 1 GPU-hour (LoRA) to 500 GPU-hours (full fine-tune). Under a moderate distribution, aggregate derivative compute reaches 0.5×–3× the base training cost of 7.7M GPU-hours. Without visibility (Scenario A), redundant derivatives accumulate freely. With DIA reporting (Scenario B), practitioners can check whether an equivalent derivative exists before creating one, plausibly reducing redundant compute by 15–30%. With DIA integrated into community norms (Scenario C), reductions of 30–50% are achievable, which is similar to the impact observed when conference reproducibility checklists reduced unreported experimental details. This pathway requires no regulatory action, it builds primarily on existing tooling, platforms, and community governance to make environmental impacts measurable and comparable at scale.

# 5. Alternative Views

## 5.1. Efficiency gains will eventually outpace demand

**Argument:** Continued improvements in quantization, distillation, and hardware efficiency will reduce AI's total footprint without requiring ecosystem-level coordination. Prior work documents declines in per-inference energy costs due to hardware and systems optimizations (Desislavov et al., 2023), and hyperscale operators have improved data centre efficiency through better PUE (Patterson et al., 2021). Proponents of this view argue that the AI industry is still in an early, high-growth phase, and that as the technology matures, efficiency gains will naturally dominate, much as they did for earlier computing paradigms.

**Response:** Efficiency gains reduce *per-run* cost, but they do not reliably reduce the *aggregate* footprint in a rapidly expanding ecosystem. As training and inference become cheaper, experimentation scales up, driving growth in training runs, fine-tuning activity, and deployments. This rebound effect is well documented in energy economics and implies that efficiency alone cannot guarantee absolute reductions (Greening et al., 2000). Historical experience shows a similar pattern in other sectors, where efficiency gains have often been offset by rising demand (Dhakal et al., 2022). The IEA projects rapid growth in electricity demand from AI and data centres under multiple scenarios, highlighting the need for measurement and coordination alongside technical progress (IEA, 2025). DIA provides this complement: a lightweight transparency layer that helps the open ecosystem translate efficiency advances into

ecosystem-level reductions rather than increased demand.

## 5.2. Reporting requirements will burden small players and suppress innovation

**Argument:** Requiring developers to track and report energy consumption will create barriers for independent researchers and startups who lack the resources and infrastructure to implement measurement systems. Compliance costs tend to fall disproportionately on smaller organizations, potentially concentrating AI development among well resourced labs that can absorb reporting overhead (Fung et al., 2007). Even voluntary standards can evolve into de facto requirements when conferences, funders, or platforms adopt them as norms, effectively raising the barrier to participation. Critics argue that the open-source ecosystem thrives precisely because of low friction contribution, and that adding accountability infrastructure risks undermining the accessibility that makes open source valuable.

**Response:** Reporting is voluntary, and coarse estimates are acceptable. In practice, transparency can help smaller players. For example, hyperscale AI developers and cloud operators such as Google, Microsoft, and Meta routinely measure and optimize data centre efficiency (e.g., PUE and WUE) for cost and capacity planning, and some publish these infrastructure-level metrics in sustainability reporting (Google Data Centers, 2025; Microsoft Datacenters, 2025; Meta, 2025). However, model-level training and inference footprints are rarely disclosed in a consistent, comparable format across releases (Henderson et al., 2020). Lightweight disclosure enables smaller teams to learn what works without repeating costly experiments. Individual-level measurement tools such as CodeCarbon and Carbontracker (Anthony et al., 2020) are also available, however, widespread reporting norms have not emerged. This is not a failure of the tools themselves but a coordination problem. We also note that if reporting norms eventually become expected or required (e.g., in conference submissions or model development), they should be introduced gradually.

## 5.3. AI's climate impact is lower than other sectors

**Argument:** Data centres account for ~1.5% of global electricity consumption, a fraction of the emissions from transportation (~23%) and heavy industry (~30%) (International Energy Agency, 2024). Critics argue that focusing policy attention and coordination resources on AI sustainability diverts effort from sectors where interventions would yield far greater absolute reductions. Climate policy literature emphasizes the importance of prioritizing high-impact sectors to maximize mitigation outcomes under constrained resources (Pacala & Socolow, 2004). From this perspective, ecosystem-level coordination mechanisms for AI may represent a misallocation of limited attention in the broader

climate policy landscape, particularly when the AI sector's relative contribution remains small.

**Response:** When smaller industries defer climate action until larger sectors move first, the cumulative effect creates a moral hazard. Coordinated impact reporting in one sector does not preclude targeted reduction efforts in another. Additionally, the current share alone is an incomplete metric because growth rate and infrastructure lock-in determine future impact. AI workloads are among the fastest-growing drivers of data centre demand; for example, AI-specific servers are growing at 30% annually, compared to 15% for data centres overall (IEA, 2025). Early interventions are typically cheaper than retrofitting once procurement, deployment habits, and software ecosystems have scaled. DIA is a low-cost mechanism to improve visibility and guide scaling decisions early, when the ecosystem is still malleable. As AI adoption grows in larger industries, greater carbon emissions in those sectors can be prevented with a framework like DIA. Importantly, water impacts are local and can be severe even when global carbon shares appear modest.

## 5.4. Reporting will be inaccurate or gamed

**Argument:** Without independent verification or auditing mechanisms, developers face incentives to underreport emissions to appear more environmentally responsible or to avoid scrutiny. Self-reported sustainability data in other domains has been widely criticized for greenwashing and selective disclosure. Research on corporate environmental reporting finds that voluntary disclosures are often incomplete, inconsistent, and strategically framed to present organizations favourably (Lyon & Maxwell, 2011; Marquis et al., 2016). If DIA relies entirely on unverified self-reporting, aggregate figures may systematically underestimate true impacts, creating a misleading picture of ecosystem-level sustainability that could delay more effective interventions.

**Response:** Imperfect reporting is better than no reporting. The primary goal of DIA is directional insight: estimating orders of magnitude, comparing alternatives, and tracking trends over time rather than auditing individual projects with high precision. Even coarse disclosure enables aggregation and anomaly detection at the ecosystem level. Community norms and reputational incentives create pressure for reasonable transparency, while automated tooling makes gross underreporting harder (e.g., hardware- and time-based estimates provide a sanity-check baseline).

## 5.5. Voluntary transparency cannot overcome economic incentives

**Argument:** The economic forces driving AI scaling, including competitive pressure, reduced marginal costs, and expanding use cases, are strong to be counteracted by voluntary disclosure. Market dynamics reward rapid scaling, and

firms that unilaterally constrain their compute usage risk falling behind competitors who do not (Tirole, 1988). Historical evidence from other domains suggests that voluntary environmental initiatives often fail to achieve meaningful reductions absent regulatory pressure. For example, studies of voluntary carbon disclosure programs find limited impact on actual emissions, with participation often serving symbolic or reputational purposes rather than driving substantive change (Delmas & Montes-Sancho, 2010).

**Response:** We agree that transparency alone is insufficient, but necessary. DIA is foundational infrastructure, not a complete solution. Transparency enables action in three ways. First, visibility creates reputational incentives: public environmental disclosure can shift behavior even without regulation (Matsumura et al., 2014). In open-source communities, visible efficiency metrics (e.g., energy per token, training kWh, or carbon intensity) can similarly encourage developers to optimize compute and environmental footprint. Second, comparability makes efficiency a practical selection criterion alongside accuracy. Third, measurement precedes management: systems such as nutritional labels and fuel economy ratings evolved from transparency to standards and, eventually, policy (Fung et al., 2007). If rebound effects are severe, complementary mechanisms (e.g., compute budgets) may be required. DIA provides the measurement foundation that such interventions depend on.

# 6. Call to Action and Implementation Path

We call on the ML community to take concrete steps toward ecosystem-level sustainability:

**For researchers and practitioners:** Include emissions and water estimates in model cards and paper submissions. Use tools (Table A3) to measure training costs and estimate inference footprints. When fine-tuning or adapting models, document the base model used and the incremental compute required. Imperfect estimates are better than none. **For conference organizers and reviewers:** Encourage environmental reporting in reproducibility checklists, with graduated expectations that distinguish resource-intensive submissions from lightweight contributions. Recognize efficiency as a first-class contribution, not merely a secondary consideration. Consider environmental impact when evaluating the significance of scaling-focused work. **For model hub operators:** Implement standardized metadata fields for carbon and water reporting. Develop dashboards that aggregate reported data across model families and their derivatives. Surface efficiency metrics alongside accuracy benchmarks in model discovery interfaces. **For cloud providers and hardware vendors:** Expose per-job carbon intensity and water usage through standardized APIs. Provide users with actionable data on the environmental cost of their workloads and enable carbon-aware scheduling by default. **For fund-**

**ing agencies:** Require environmental impact statements in grant proposals for compute-intensive research. Consider efficiency and sustainability as evaluation criteria alongside scientific merit. **For open-source labs and foundations:** Lead by example with comprehensive environmental reporting for flagship releases. Invest in tooling that reduces reporting friction. Participate in developing community standards for sustainability accounting. **For downstream deployers:** Report inference-phase footprints, which often dominate lifecycle emissions for widely used models. Adopt efficiency benchmarks alongside performance metrics in procurement and deployment decisions.

Earth does not distinguish between emissions from open and closed-source models, between base models and derivatives, between training and inference. The infrastructure we build today will determine whether open-source AI develops responsibly or experiences uncoordinated growth. We believe it can achieve the former. The same community that built transformers, democratized LLMs, and showed that collaborative development can compete with corporate R&D can also coordinate on sustainability.

**Complementarity to broader governance.** Efficiency alone cannot deliver sustainable AI; governance is also necessary. We emphasize that DIA is *non-regulatory* and does not restrict who can train, fine-tune, or release models. However, historical experience across domains suggests that large-scale risk reduction rarely emerges from voluntary action alone. Public health and safety improvements have often depended on shared standards, disclosure norms, and institutional mechanisms, e.g., workplace smoke-free policies (Meyers et al., 2009), fire safety codes (Hall et al., 2022), and seatbelt laws (CDC, 2020) that translated best practices into population-level outcomes. DIA should be understood as a foundational transparency infrastructure, a minimal coordination layer that can operate within open-source ecosystems today while supporting stakeholders (conferences, funders, procurement teams, policymakers).

**A path forward.** We propose that by 2027, major open-source model releases include standardized DIA reports covering training emissions, water usage, and documented lineage. Achieving this requires no regulatory mandate (only coordination). The tools exist; the data can be collected; the community has demonstrated its capacity for collective action. What remains is the decision to act.

## 7. Related Work

Recent position papers reinforce complementary aspects of our argument. One work (Wilder & Zhou, 2026) emphasizes that AI evaluation standards should extend beyond traditional performance metrics, aligning with DIA's inclusion of environmental cost alongside accuracy. Similarly,

(Upadhyay et al., 2026) advocates for releasing smaller analog models to reduce downstream compute demands; DIA provides the necessary accounting framework to assess whether such interventions yield net efficiency gains. More broadly, (McCoy et al., 2026) argues for measuring AI progress in terms of capability per resource, rather than scale alone, directly supporting our call for efficiency-aware reporting. Our proposal connects three research traditions: (i) ML carbon and water measurement, (ii) commons governance theory, and (iii) sustainability reporting frameworks. DIA sits at their intersection as a coordination layer.

**Carbon Footprinting and Efficiency in ML.** Early work showed that training large NLP models can incur substantial $CO_2$ emissions (Strubell et al., 2019), motivating the Green AI agenda and calls to treat efficiency as a first-class objective (Schwartz et al., 2020). Subsequent studies introduced practical tools for estimating and tracking emissions (e.g., ML $CO_2$ Impact, CodeCarbon, Carbontracker; Table A3) while also highlighting persistent challenges, including methodological inconsistency and missing metadata. These efforts primarily support *job- or model-level* footprinting. In contrast, DIA treats footprint reporting as *ecosystem-level infrastructure*.

**Commons Governance Theory.** Our framing draws on Ostrom's theory of commons governance (Ostrom, 1990), which emphasizes that shared resources can be sustainably managed through self-organized institutions, monitoring, and collective norms, rather than inevitable collapse under the "tragedy of the commons" (Hardin, 1968). This perspective has been extended to digital and knowledge commons (Hess & Ostrom, 2007) and to open-source software ecosystems using the Institutional Analysis and Development framework (Schweik & English, 2012; 2013). Open-source AI resembles a commons because participation is open, but environmental externalities accumulate across the ecosystem. Unlike traditional open-source software, the key risk is not abandonment but uncontrolled aggregate resource use through repeated training, fine-tuning, and deployment. DIA operationalizes a commons-oriented response by providing visibility and lightweight monitoring at the ecosystem level without restricting access.

**Regulatory Frameworks and Voluntary Standards.** AI sustainability reporting is emerging but not standardized. The EU AI Act requires providers of general-purpose AI models to document known or estimated energy consumption and encourages voluntary codes of conduct on environmental sustainability (European Parliament and Council, 2024; Pagallo, 2025). Voluntary standards such as the Green Software Foundation's Software Carbon Intensity provide methods for measuring software emissions (Green Software Foundation, 2024). DIA complements these approaches by supporting bottom-up coordination in open ecosystems.

# 8. Conclusion

Open-source AI has democratized access to powerful models, but its success has also created a coordination gap where the cumulative footprint of countless derivatives remains largely invisible. While efficiency gains are valuable, they cannot reliably reduce aggregate impact under rapid growth and rebound effects. To address this, we propose **Data and Impact Accounting** as a lightweight, non-regulatory transparency layer that standardizes footprint reporting and enables ecosystem-level aggregation. By making carbon and water impacts measurable and comparable across model lineages, DIA helps the open-source community identify hotspots, benchmark progress, and make better-informed decisions about when and how to scale open-source AI.

*Impact:* DIA provides essential measurement to support interventions such as compute budgets when rebound effects are significant. By making carbon and water costs visible, DIA could help reduce aggregate emissions across the ecosystem. However, voluntary reporting may not overcome incentives driving AI scaling, and such frameworks can encourage optimizing metrics rather than real environmental benefit. We do not address supply chain impacts or restrict open-source access.

# Acknowledgements

Resources used in preparing this research were provided, in part, by the Province of Ontario, the Government of Canada through CIFAR, and companies sponsoring the Vector Institute http://www.vectorinstitute.ai/#partners. GWT acknowledges support from the Natural Sciences and Engineering Research Council (NSERC), the Canada Research Chairs program, and the Canadian Institute for Advanced Research (CIFAR) Canada CIFAR AI Chairs program.

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

# Appendix

## A. Formulae

### A.1. Energy Consumption

When direct energy measurements are unavailable, we estimate electricity consumption from aggregate GPU compute time:

$$E_{\text{train}} = \frac{H_{\text{GPU}} \times P_{\text{avg}} \times \text{PUE}}{1000}, \quad (1)$$

where $E_{\text{train}}$ is total energy in kWh, $H_{\text{GPU}}$ is aggregate GPU-hours across all devices, $P_{\text{avg}}$ is average GPU power draw, and PUE is power usage effectiveness. When measured power is unavailable, we approximate $P_{\text{avg}}$ using vendor TDP values (see Table 1, footnote h). Since actual power draw typically ranges from 60–80% of TDP depending on utilization, this provides an upper-bound estimate of energy consumption.

Table A1. GPU hardware assumptions for energy estimation, based on vendor specifications (NVIDIA Corporation, 2017; 2020; 2022).

| Era | GPU | TDP (W) | Models |
|---|---|---|---|
| 2020 | V100 | 300 | GPT-3 |
| 2022–2023 | A100-80GB | 400 | OPT, BLOOM, Falcon, Llama 2 |
| 2024 | H100-80GB | 700 | LLaMA 3/3.1 |
| 2024 | H800 | 350 | DeepSeek-V3 |

### A.2. Carbon Emissions

Carbon emissions are computed as :

$$C_{\text{train}} = \frac{E_{\text{train}} \times \text{CI}}{1000}, \quad (2)$$

where $C_{\text{train}}$ is in tCO2eq and $CI$ is grid carbon intensity in kgCO2/kWh.

### A.3. Water Consumption

Water consumption is estimated as :

$$W_{\text{train}} = E_{\text{train}} \times \text{WUE}_{\text{total}}, \quad (3)$$

where $W_{\text{train}}$ is in litres and $\text{WUE}_{\text{total}}$ (L/kWh) combines on-site cooling and off-site electricity generation water *consumption* (i.e., water evaporated or otherwise not returned to local sources). When reporting in megalitres (ML), we use $W_{\text{train}}^{(\text{ML})} = W_{\text{train}}/10^6$.

## B. Table 1 Notes

[a] **Model sources:** GPT-3 (Brown et al., 2020); BLOOM (Workshop et al., 2022); OPT (Zhang et al., 2022); Falcon 180B (Almazrouei et al., 2023); Llama 2 (Touvron et al., 2023); Llama 3 (Dubey et al., 2024);

Table A2. Typical parameter ranges used for estimation when not reported.

| Parameter | Typical range | Unit |
|---|---|---|
| PUE (hyperscale) | 1.1–1.2 | – |
| Carbon intensity ($CI$) | 0.1–0.6 | kgCO2/kWh |
| WUE_total | 1.8–4.0 | L/kWh |
| Tree absorption | ∼25 | kgCO2/tree/year |

Llama 3.1 (Meta, 2024); Mistral 7B (Jiang et al., 2023); GPT-4 (Achiam et al., 2023); DeepSeek-V3 (Liu et al., 2024).

[b] **tCO2eq sources:** GPT-3 (Patterson et al., 2021); BLOOM (Luccioni et al., 2023); OPT (Zhang et al., 2022); Llama 2 (Touvron et al., 2023); Llama 3 (Dubey et al., 2024); Llama 3.1 (Meta, 2024); GPT-4 range derived from IEA energy estimate (International Energy Agency, 2024) with carbon intensity 0.1–0.445 kgCO2/kWh; Falcon and DeepSeek derived from disclosed GPU-hours (see [d], [g]).

[c] **Water estimation:** Water values are estimated using a representative WUE_total range of 1.8–4.0 L/kWh, intended to capture combined on-site cooling and off-site electricity-generation water consumption. These are order-of-magnitude estimates and vary substantially by region and facility configuration.

[d] **Falcon 180B:** Estimated from 7 M A100 GPU-hours at 400 W average draw, PUE 1.1, grid intensity 0.39 kgCO2/kWh.

[e] **GPT-4:** Range based on IEA's 42.4 GWh estimate (International Energy Agency, 2024) with carbon intensity 0.1–0.445 kgCO2/kWh.

[g] **DeepSeek-V3:** Estimated from 2.79 M H800 GPU-hours at 350 W average draw (assuming 50% average utilization), PUE 1.1, grid intensity 0.51 kgCO2/kWh.

[h] **Hardware assumptions:** When not reported, we use vendor *thermal design power* (TDP) values as upper bounds (V100 300 W; A100-80GB 400 W; H100-80GB 700 W), with typical utilization 60–80% depending on workload (Henderson et al., 2020; Dodge et al., 2022).

## C. Environmental measurement tools and metrics

The environmental measurement tools and metrics are given in Table A3.

## D. Carbon Emission Estimation

Following established methodology (Strubell et al., 2019; Patterson et al., 2021), we estimate training-related carbon emissions using:

$$\text{CO}_2\text{eq} = N \times P \times T \times \text{PUE} \times \text{CI} \quad (4)$$

where each variable is defined in Table A4.

*Table A3.* Environmental measurement tools and metrics.

| Resource | Tool/Metric | Type | Description |
|---|---|---|---|
| **Carbon** | ML $CO_2$ Impact (Lacoste et al., 2019) | Est. | Job-level emissions estimator |
| | CodeCarbon (Courty et al., 2024) | Track | Automated energy/emissions tracking |
| | Carbontracker (Anthony et al., 2020) | Track | Energy/carbon tracking with prediction |
| | NVML/nvidia-smi (NVIDIA Corporation, 2024) | Meas. | GPU power monitoring |
| | Kepler (CNCF, 2024) | Meas. | K8s pod/node power metrics |
| | Cloud Carbon Footprint (Thoughtworks, 2024) | Est. | Multi-cloud emissions dashboards |
| | Electricity Maps (Electricity Maps, 2024) | Data | Real-time carbon intensity |
| | Carbon Aware SDK (Green Software Foundation, 2024) | Sched. | Emission-aware scheduling |
| **Water** | WUE (The Green Grid, 2011) | Metric | Water Usage Effectiveness (L/kWh) |
| | Google Env. Reports (Google, 2024) | Data | Facility-level water disclosure |
| | Aqueduct Atlas (World Resources Institute, 2024) | Data | Global water stress mapping |
| | Li et al. (Li et al., 2025) | Method | AI water footprint estimation |
| | EU Reg. 2024/1364 (European Commission, 2024) | Reg. | Mandatory WUE reporting (EU) |

*Table A4.* Variables for carbon emission estimation with representative values for GPT-4.

| Variable | Definition | Value | Source |
|---|---|---|---|
| $N$ | Number of accelerators | 25,000 GPUs | (Ludvigsen, 2023) |
| $P$ | Power per accelerator | 0.4 kW | (Patterson et al., 2021) |
| $T$ | Training duration | 2,400 h (100 days) | (Ludvigsen, 2023) |
| PUE | Power Usage Effectiveness | 1.2 | (IEA, 2025) |
| CI | Carbon intensity | 0.4 $kgCO_2$/kWh | (IEA, 2025) |

Substituting values from Table A4 into Equation 4:

$$
\begin{aligned}
CO_2eq &= N \times P \times T \times PUE \times CI, \\
&= 25,000 \times 0.4\,\text{kW} \times 2,400\,\text{h} \times 1.2 \times 0.4\,kgCO_2/\text{kWh} \\
&= 11,520,000\,kgCO_2eq \\
&\approx \boxed{11,520\,tCO_2eq}
\end{aligned}
\tag{5}
$$

This estimate aligns with published ranges of 10,000–15,000 $tCO_2$eq (Patterson et al., 2021; Ludvigsen, 2023).

### D.1. Worked example

DeepSeek-V3 reports 2.788M (2.79 M) H800 GPU-hours for training (Liu et al., 2024).Since measured power draw is unavailable, we estimate using vendor TDP and assumed utilization:

$$
P_{\text{avg}} = \text{TDP} \times \text{utilization} = 700\,\text{W} \times 0.50 = 350\,\text{W} \tag{6}
$$

**Step 1: Energy.**

$$
\begin{aligned}
E_{\text{train}} &= \frac{H_{\text{GPU}} \times P_{\text{avg}} \times \text{PUE}}{1000} \\
&= \frac{2,790,000 \times 350 \times 1.1}{1000} \\
&= 1,074,150\,\text{kWh} \approx 1.07\,\text{GWh}
\end{aligned}
\tag{7}
$$

**Step 2: Carbon emissions.** Using China's average grid carbon intensity of 0.51 $kgCO_2$/kWh:

$$
C_{\text{train}} = \frac{E_{\text{train}} \times \text{CI}}{1000} = \frac{1,074,150 \times 0.51}{1000} \approx 548\,tCO_2eq \tag{8}
$$

**Step 3: Water consumption.** Using $WUE_{\text{total}}$ range of 1.8 - 4.0 L/kWh:

$$
\begin{aligned}
W_{\text{low}} &= 1,074,150 \times 1.8 = 1,933,470\,\text{L} \approx 1.9\,\text{ML} \\
W_{\text{high}} &= 1,074,150 \times 4.0 = 4,296,600\,\text{L} \approx 4.3\,\text{ML}
\end{aligned}
\tag{9}
$$

These estimates are consistent with the values reported in Table 1 ($\sim$545 $tCO_2$eq, 1.9–4.3 ML).

**Assumptions and limitations.** The H800 GPU has a TDP of 700 W; we assume 50% average utilization based on the mixed-expert architecture of DeepSeek-V3, which activates a subset of parameters per token. PUE is set to 1.1 (typical for hyperscale facilities). The CI value of 0.51 $kgCO_2$/kWh reflects China's national average grid; regional variation (e.g., Guizhou hydropower regions) could lower this substantially. Estimated carbon emissions depend on three key parameters: GPU utilization (affecting $P_{\text{avg}}$), PUE, and grid carbon intensity (CI). Table A5 shows how the GPT-4 and DeepSeek-V3 estimates vary under different assumptions.

*Table A5.* Sensitivity of carbon emission estimates ($tCO_2$eq).

| Scenario | Utilization | PUE | CI (kgCO₂/kWh) |
|---|---|---|---|
| GPT-4 (25,000 GPUs × 2,400 h) | | | |
| Conservative | 60% | 1.1 | 0.1 → 1,584 tCO₂eq |
| Mid estimate | 80% | 1.2 | 0.4 → 9,216 tCO₂eq |
| Upper bound | 100% | 1.2 | 0.445 → 12,816 tCO₂eq |
| **DeepSeek-V3** (2.79M GPU-h) | | | |
| Conservative | 40% | 1.1 | 0.35 → 301 tCO₂eq |
| Mid estimate | 50% | 1.1 | 0.51 → 548 tCO₂eq |
| Upper bound | 70% | 1.2 | 0.6 → 984 tCO₂eq |

The sensitivity analysis reveals that estimates can vary by a factor of $\sim$3–8$\times$ depending on assumptions. For GPT-4, the existing estimate of 11,520 $tCO_2$eq from Equation (5) falls within the mid-to-upper range of this sensitivity analysis. The largest source of uncertainty is grid carbon intensity, which varies from $\sim$0.1 $kgCO_2$/kWh (hydropower-dominated grids) to $>$0.6 $kgCO_2$/kWh (coal-heavy grids). GPU utilization is the second-largest factor, as actual power draw during training depends on workload characteristics,

batch sizes, and memory access patterns. These ranges underscore the importance of standardized reporting (as proposed by DIA) to reduce reliance on third-party estimation.

*Table A6.* Reporting gap audit: disclosure of DIA fields across 10 Hugging Face model releases (5 base models, 5 derivatives). Water consumption is unreported across all models. Derivatives disclose no environmental fields beyond lineage.[1]

| DIA Field | Base (n=5) | Deriv. (n=5) | Overall |
|---|---|---|---|
| GPU-hours | 4/5 (80%) | 0/5 (0%) | 4/10 (40%) |
| Energy (kWh) | 1/5 (20%) | 0/5 (0%) | 1/10 (10%) |
| $CO_2$ (tCO$_2$eq) | 4/5 (80%) | 0/5 (0%) | 4/10 (40%) |
| Water (L) | 0/5 (0%) | 0/5 (0%) | 0/10 (0%) |
| Lineage | 5/5 (100%) | 5/5 (100%) | 10/10 (100%) |

### D.2. Example DIA Report

To illustrate how DIA reporting works in practice, we present a concrete example: fine-tuning Llama 3-8B using QLoRA on a single A100-80GB GPU for 4 hours on an AWS `us-east-1` instance.

**Step 1: Compute the footprint.** Using the formulae from Appendix A:

- **Energy:** $E = 1 \times 400\,\text{W} \times 4\,\text{h} \times 1.1\,(\text{PUE})/1000 = 1.76$ kWh

- **Carbon:** $C = 1.76 \times 0.4\,(\text{CI})/1 = 0.70$ kgCO$_2$eq

- **Water:** $W = 1.76 \times [1.8, 4.0]\,(\text{WUE}) = 3.2$–$7.0$ litres

**Step 2: Attach to model metadata.** The following YAML block can be added directly to a Hugging Face model card or repository `README.md`:

```
dia_report:
  base_model: meta-llama/Llama-3-8B
  method: QLoRA
  hardware:
    gpu: A100-80GB
    count: 1
  duration_gpu_hours: 4.0
  energy_kwh: 1.76
  carbon_kgco2eq: 0.70
  water_liters: 3.2-7.0
  region: us-east-1
```

```
  carbon_intensity_kgco2_per_kwh: 0.4
  wue_l_per_kwh: 1.8-4.0
  tool: codecarbon-2.4.1
  data_quality:
    energy: measured
    carbon: estimated-from-region
    water: estimated-from-default-wue
```

**Step 3: Paper reporting.** For a conference submission, the same information can appear in a reproducibility or ethics checklist as a single line:

```
Training: 1x A100, 4 GPU-h, 1.76 kWh,
0.70 kgCO2eq, 3.2-7.0 L water (us-east-1).
Base model: Llama-3-8B. Tool: CodeCarbon.
```

The `data_quality` field is critical: it distinguishes measured values (e.g., energy from CodeCarbon) from estimates (e.g., water from default WUE ranges). This allows downstream aggregation tools to weight or filter by confidence level. The `base_model` field enables lineage tracking, so ecosystem dashboards can sum the footprint of a model family across all its derivatives. The schema is intentionally minimal — a practitioner can fill it in under five minutes using information already available from standard training logs.

---

[1]Base models audited: `meta-llama/Meta-Llama-3-8B`, `meta-llama/Llama-3.1-405B`, `meta-llama/Llama-3.2-1B`, `bigscience/bloom`, `mistralai/Mistral-7B-v0.1`. Derivatives audited: `Gradient/Llama-3-8B-Instruct-262k`, `LoneStriker/Meta-Llama-3-70B-Instruct-GGUF`, `NousResearch/Meta-Llama-3-8B`, `meta-llama/Llama-3.2-3B-SpinQuant`, `QuantFactory/Llama-3-8B-Instruct-262k-GGUF`. All model cards accessed March 2026.

