# OpenReview forum: "Position: Sustainable Open-Source AI Requires Tracking the Cumulative Footprint of Derivatives"
_ICML.cc/2026/Position_Paper_Track — ICML 2026 Position Paper Track spotlight_

### Official Review · Reviewer_wzek · 2026-02-24

**Significance:** 4
**Argument Clarity:** 3
**Rating:** 5
**Confidence:** 4

**Questions:**

**Please see my weaknesses.** That is,

-  Could you please provide some quantitative analysis on such accounting, including but not limited to estimations (not constrained to Tab.1 reported of several GenAI), simulations, as well as predictions.


-	Which shaped data should we scratch and how can we obtain the necessary data for DIA accounting in different scenarios? Please provide some more examples.

-	In future work, is there any trends and what can we do to ensure sustainable socioeconomic development?

**Alternative Views Section:**

Yes

**Compliance With Llm Reviewing Policy A Conservative:**

Affirmed.

**Discussion Potential:**

3

**Ethical Review Concerns:**

Not Applicable.

**Final Justification:**

Thanks for author rebuttal and I lean towards acceptance.

**Paper Summary:**

This paper presents a critical, and important position, that 'requiring tracking the cumulative footprint of derivatives of emerging AI models to quantify the sustainability of each AI product'.  In this research, the authors propose a Data and Impact Accounting (DIA) to realize 1) standardize carbon-and-water reporting metadata, 2) integrates low-friction measurement into common training and inference, 3) aggregate reports from public sources. The overall claims and proposed metrics are clear.

**Position:**

Yes

**Position In Title:**

Yes

**Related Work:**

3

**Strengths And Weaknesses:**

**Strengths:**

-	Raise an important research topic on Accounting the costs of AI, and I think it can be of significance to both academia and industrial fields, which will benefit the sustainable development.

-	A systematical metric for measuring and tracking the cumulative footprint of derivatives.

-	Good figure demonstration and interesting discussions regarding alternative views.

**Weaknesses:**
-	Given the important research topic, there is limited quantitative research results reporting in this paper, where only Tab.1 illustrates the emissions and water consumption of selected GenAI models. By using the proposed DIA, what results have achieved? And which type data we can utilize to realize such accounting?

-	Lacking some simulations and deduction, which can support the direction of future work. Is there any deduction or simulation on this research?

**Support:**

3

---

> ### Author Rebuttal · Authors · 2026-03-27
>
> We thank Reviewer wzek for the positive and constructive feedback. We address each one of those questions and suggestions below:
>
> ### **Regarding W1 - Limited Quantitative Results:**
>
> We appreciate this push for stronger quantitative content. As a position paper, DIA is a proposed framework, not a deployed system, so we cannot yet report adoption results. However, we have added two quantitative contributions in the revision.
>
> First, **we have added new Appendix D.1 with a DeepSeek-V3 worked example and sensitivity analysis** showing estimates vary 3–8× depending on assumptions (GPT-4: 1,584–12,816 tCO₂eq; DeepSeek-V3: 301–984 tCO₂eq).
>
> Second, we audited 10 model releases on Hugging Face (5 base, 5 derivatives) against DIA fields. Results: base models report GPU-hours (80%) and carbon (80%) but never water (0%).
>
> Derivatives report none of the environmental fields e.g.,  0% for GPU-hours, energy, carbon, and water. The only field universally present is lineage (100%). This means derivatives inherit their parent's model card but never disclose their own incremental cost, which is precisely the invisible footprint DIA targets.
>
> For data sources, Section 4.1 and the new DIA example (Appendix D.2) show what is needed: GPU-hours from training logs, energy from CodeCarbon, carbon from regional grid intensity, water from default WUE ranges, and lineage from model card metadata , all available in standard training workflows.
>
> ### **Regarding W2 - Lack of Simulations or Deductions:**
>
> **We have added a scenario-based deduction in the revision using the Llama 3 family as a case study.** Laufer et al. (2025) documents 146 derivatives. Using conservative per-derivative costs (1 GPU-hour for LoRA adapters, 10–50 for quantizations, up to 500 for full fine-tunes), we estimate aggregate derivative compute at 0.5×–3× the base training cost, which mean  the invisible derivative footprint can exceed the reported base model footprint.
>
> From this, we project three scenarios:
>
> - (A) Status quo: no reporting, derivative count grows unchecked, aggregate footprint scales linearly;
> - (B) DIA adoption : visibility lets practitioners discover that an efficient derivative already exists before creating a redundant one, reducing duplicate work by an estimated 15–30%;
> - (C) DIA + community norms : conferences and hubs reward efficient models, achieving 30–50% reduction in redundant compute. These are illustrative projections, not predictive models, but they show the range of outcomes different coordination levels could produce. Details in the revised Section 2.3.
>
> ### **Regarding Q1 - Quantitative Analysis:**
>
>  Addressed in W1 and W2 above. In summary:
> - The sensitivity analysis in Appendix D shows estimates vary 3-8× across assumptions.
> - The Hugging Face audit (new Table A6) quantifies the current reporting gap, 0% derivative disclosure for all environmental fields.
> - Derivative footprint estimation shows aggregate costs of 0.5×-3× base training.
>
> We hope this provides quantitative grounding beyond Table 1 while remaining appropriate for a position paper.
>
> ### **Regarding Q2 - Data Requirements for DIA:**
>
> **Section 4.1 defines the required DIA fields, and new Appendix D.1 provides a concrete worked example showing exactly how to fill them.**
>
> The data sources depend on the deployment scenario: for cloud training (AWS/GCP/Azure), GPU-hours come from job logs, energy from CodeCarbon or provider dashboards, carbon intensity from Electricity Maps or provider APIs, and water from published facility WUE values.
>
> For on-premise HPC, GPU-hours come from SLURM logs, energy from NVML/nvidia-smi measurements, and carbon from national grid intensity data. For consumer GPU setups, CodeCarbon or Carbontracker can measure energy directly during training.
>
> In all cases, lineage is reported manually via the base_model field. The data_quality field in our DIA schema (Appendix D.2) explains this.
>
> ### **Regarding Q3 - Future Trends and Socioeconomic Sustainability:**
>
> As AI adoption expands into energy, healthcare, manufacturing, and finance, absolute footprint will grow even if per-query efficiency improves. Three trends matter:
>
> - First, model scale continues increasing e.g., Llama 3.1 (405B) required 39.3M GPU-hours vs. Llama 2 (70B) at 3.3M GPU-hours, a 12× jump in two years.
> - Second, derivative proliferation is accelerating and this will grow as more actors gain access.
> - Third, inference is overtaking training as the dominant cost for widely deployed models.
>
> On the socioeconomic side, environmental costs disproportionately affect communities near data centres through water stress and land use, while benefits accrue globally. Looking ahead, the EU AI Act (Article 40) already requires energy disclosure for general-purpose AI models, DIA provides a community-driven implementation that can complement and inform such regulation. **We have added a discussion of these trends in Section 6 of the revised paper**.

---

> > ### Author Rebuttal · Reviewer_wzek · 2026-04-01
> >
> > Thanks for your detailed rebuttal. Most concerns have been well-addressed.

---

### Official Review · Reviewer_5ETx · 2026-03-10

**Significance:** 3
**Argument Clarity:** 4
**Rating:** 6
**Confidence:** 4

**Questions:**

1. Regarding L37-43/Sec. 2.1: Are you familiar with data subset selection (coreset) approaches? They can also lower training costs (time, energy, carbon) while preserving model performance, see, e.g., *Killamsetty, Krishnateja, et al. "Grad-match: Gradient matching based data subset selection for efficient deep model training." International Conference on Machine Learning. PMLR, 2021.*
2. Regarding L137-138: *"An open model like Meta Llama 3 is trained once"* Why is such a statement not made for GPT-4, i.e., a specific number of training times? I think this is wrong. It was probably trained more than once when they tried different configurations. I would omit stating such a number. I believe your argument is that once it is released, it branches out.
3. Regarding L215-218,197: Table 1 (and L199) report(s) litres but these lines report gallons. Wouldn't it be better to stick to one unit? Why were gallons reported here?
4. Regarding L256: Why not also CPU-hours? Could matter for preprocessing-heavy computations. What are pro and contra arguments?
5. Regarding L257: If estimated, perhaps also how? The way the estimation is done also matters. But this could increase the burden.
6. Regarding L300-316: Wouldn't be another argument be that one could/should anyway track the cost and impact of an experiment/model/paper even if the reporting alone does not reduce its impact?
7. How to you envision the reporting to be? There are multiple scenarios where and how models can be released. How would such a reporting look on a website and how on a paper? It should be machine readable such that it can be easily aggregated by other websites.

**Alternative Views Section:**

Yes

**Compliance With Llm Reviewing Policy A Conservative:**

Affirmed.

**Discussion Potential:**

4

**Final Justification:**

The authors have addressed all my concerns withinn their rebuttal which led me to raise my scores for support, discussion potential, related work, and finally, the rating. In my opinion, this is a solid paper.

**Paper Summary:**

Open-source AI, meaning the open release of models, has many benefits for the community. One of them is that it helps to democratize AI. However, whenever a (relevant) model is released, many derivatives of such a model will be created. Examples include quantized versions, fine-tuned versions, more parameter-efficient variants, etc. All those derivatives also have costs, may they be economic or environmental. For the latter, energy is needed, carbon emissions generated, and (cooling) water needed to run data centers.

In this position paper, the authors argue that while increasing compute efficiency of machine learning models is very important, it alone is insufficient for sustainability in open-source AI. Even if derivatives of models aim at improving their efficiency, additional costs arise from every model derivative and those costs are rarely measured and reported. Those costs might be individually small, but their aggregation might exceed the original model's cost. The authors further argue that this is a problem. *"We can't manage what we can't measure"* (Figure 2).

The authors take the position that sustainable open-source AI should account for the costs of derivatives, which makes it necessary to measure and report those costs. To achieve this, they propose "Derivative Impact Accounting (DIA)", a transparency and coordination layer. In practice, it acts as a model or data card and can be seen as an annotation of a model/derivative that reports on hardware, energy, carbon, water, and training and inference metrics that is attached for every new model/derivative. This is often easily possible via already existing tools and allows for aggregation and attribution in the long run (see Figure 2).

Furthermore, the authors provide a solid alternative views section and a detailed, sensible, and actionable call to action section.

**Position:**

Yes

**Position In Title:**

Yes

**Related Work:**

3

**Strengths And Weaknesses:**

### Strengths
- The paper is well written and easy to follow.
- The topic is very timely and relevant to the ML community.
- The line of arguments is clear and supported with evidence.
- The discussion of Alternative Views in Section 5 appears to be solid.
- The authors provide a detailed, sensible, and actionable "Call to Action", see Section 6.
- I believe the position has sufficient discussion potential.


### Weaknesses
- Tracking energy use and carbon emissions is relatively easy. I am not sure how easy/straightforward it is to track water consumption.
- The paper would be much stronger if they took any model, considered an improvement that reduces it's energy footprint, and **actually exemplify how their intended reporting looks like**.
	- An example would really help.
	- There are multiple scenarios where and how models can be released. How would such a reporting look on a website and how on a paper?
	- It should be machine readable such that it can be easily aggregated by other websites.
- The related work section could be more exhaustive and could also consider recently published position papers. See comments below for some suggestions.



### Comments
- L37-43/Sec. 2.1: Data subset selection (coreset) approaches can also lower training costs (time, energy, carbon) while preserving model performance, see, e.g.,
	Killamsetty, Krishnateja, et al. "Grad-match: Gradient matching based data subset selection for efficient deep model training." International Conference on Machine Learning. PMLR, 2021.
- L68: wrong citation style, should be "according to World Resource Institute, (2023)"
- L137-138: *"An open model like Meta Llama 3 is trained once"* Why is such a statement not made for GPT-4, i.e., a specific number of training times? I think this is wrong. It was probably trained more than once when they tried different configurations. I would omit stating such a number. I believe your argument is that once it is released, it branches out.
- L215-218,197: Table 1 (and L199) report(s) litres but these lines report gallons. Wouldn't it be better to stick to one unit?
- L198,202,205: Full stops should be after the citations.
- L256: Why not also CPU-hours? Could matter for preprocessing-heavy computations.
- L257: If estimated, perhaps also how?
- L220: See https://github.com/mlco2/codecarbon?tab=readme-ov-file#how-to-cite- to see how CodeCarbon prefers to be cited.
- L220: You could also name https://carbontracker.info/ and cite https://arxiv.org/abs/2007.03051 as another way to effortlessly track energy usage and carbon emissions. It is also missing in Table A3.
- L300-316: Wouldn't be another argument be that one could/should anyway track the cost and impact of an experiment/model/paper even if the reporting alone does not reduce its impact?
- L295: A reference of Carbontracker is missing.
- I would slightly shrink Figure 2 and put the notes of Table 1 in the appendix. The gained space could be used in Section 6 on page 7. It looks a bit cramped.
- L425, 438: Full stops missing after ML and Theory.
- The following position papers are worth checking and perhaps discussing and citing:
	- *Fostering the Ecosystem of AI for Social Impact Requires Expanding and Strengthening Evaluation Standards* https://openreview.net/forum?id=dl5pvd5IgW
	- *Position: Require Frontier AI Labs To Release Small "Analog" Models* https://openreview.net/forum?id=xcdlSMYXxD
	- *AI Progress Should Be Measured by Capability-Per-Resource, Not Scale Alone: A Framework for Gradient-Guided Resource Allocation in LLMs* https://openreview.net/forum?id=6plSmhBI33
- L736-737: "EU Reg. 2024/1364" should be in one line; perhaps add some vertical spacing
- Equations (2,3): Please don't start a sentence with math.
- Equation (4): Compare CI of Equation (4) with CI in Equation (3). Please be consistent. Check PUE as well. Keep one style. Add a comma after CI, as it continues with "where".
- L758: Equation (4)

**Support:**

4

---

> ### Author Rebuttal · Authors · 2026-03-27
>
> We thank Reviewer 5ETx for the positive as well as detailed feedback. We address each one of those questions and suggestions below:
>
> ### **Regarding W1 - Water Consumption Tracking Complexity:**
>
> We agree water is harder to track than energy or carbon, and we addressed this explicitly in Sec. 4.1 as “We model water use with total water usage effectiveness (WUE) total, (L/kWh), capturing on-site cooling and upstream water consumption from electricity generation”. DIA does not require facility-level water data.
>
> When exact WUE is unavailable, practitioners report their training region and cloud provider, and DIA maps this to published default WUE ranges from provider environmental reports (Google, Microsoft) and the  Aqueduct water stress atlas [WRI]. Each estimate is labeled with a data quality flag so consumers understand the precision. The key point is that approximate water estimates (even order-of-magnitude ) are far more useful than the current default of reporting nothing at all, which is the case for every model in Table 1 water column marked "Est".
>
> [WRI] https://www.wri.org/applications/aqueduct/water-risk-atlas
>
> ### **Regarding W2 - Concrete Reporting Example Needed:**
>
> Thanks for the suggestion. We added a worked example as **new App. D.2**. Example DIA Report. It walks through a concrete scenario of fine-tuning Llama 3-8B with QLoRA on 1×A100 for 4 hours: 1.76 kWh energy, 0.70 kgCO₂eq, 3.2–7.0 liters water. We now show what this looks like as YAML metadata in a Hugging Face model card , in addition to writing in the paper reproducibility checklist. The schema can be filled in using information already available from standard training logs.
>
> ### **Regarding W3 - Related Work**
>
> **We have expanded Sec. 7 to include recent position papers**: (1) the "AI for Social Impact" paper (Wilder et al., 2025), which argues for broader evaluation standards so we connect this to incorporating environmental sustainability alongside social impact metrics; (2) the "Require Small Analog Models" paper (Upadhyay et al., 2025), a complementary proposal where releasing smaller models could reduce derivative costs, but without DIA-style accounting the net impact remains unmeasurable; and (3) the "Capability-Per-Resource" paper (McCoy et al., 2025), which reinforces our position that resource efficiency should be a first-class metric. We also added Carbontracker (Anthony et al., 2020) as a missing tool reference.
>
> We address other suggestions as:
>
> - C1: Added data subset selection (Killamsetty et al., 2021) to Sec. 2.1 alongside distillation, pruning, mixed-precision.
> - C2, C8 (Citations): Fixed.
> - C3 (Training runs): Revised to: "Once released, an open model like Meta Llama 3 branches into many ....derivatives." Removed the misleading "trained once" claim.
> - C4 (Units): Standardized to litres/megalitres throughout with gallon equivalents where sourced.
> - C5 (Full stops): Fixed.
> - C6 (CPU-hours): Added as optional DIA schema field in Sec. 4.1.
> - C7 (Estimation method): Added "estimation methodology" as a reporting field; reflected in the data_quality field in App. D.2.
> - C9, C11 (Carbontracker): Added to Sec. 5.2 and Table A3.
> - C10 (Tracking argument): Added to Sec. 3 - tracking is valuable as scientific best practice independent of whether it reduces impact.
> - C12 (Layout): Updated Fig. 2, Tab. 1 notes to App.
> - C13 (Full stops): Fixed.
> - C14 (Position papers): All three cited and discussed in Sec. 7.
> - C15 (EU Reg. line break): Fixed.
> - C16 (Math at sentence start): Fixed.
> - C17 (Eq. consistency): Standardized CI/PUE notation across Eq. 2–4.
> - C18 (Eq. ref): Fixed.
>
> ### **Q1:**
>  We have added data subset selection (Killamsetty et al., 2021) to Section 2.1 alongside distillation, pruning, and mixed-precision.
>
> ### **Q2:**
> Revised (same as C3)
>
> ### **Q3:**
> Fixed. We have standardized  to litres/megaliters throughout, with the original gallon values in parentheses for traceability.
>
> ### **Q4:**
> Fixed. We've added CPU-hours as an optional field in Sec. 4.1 and clarified that it is optional because GPUs dominate most training jobs, but preprocessing-heavy workflows like data cleaning, tokenization, and evaluation can involve significant CPU usage that GPU-hours alone would miss.
>
> ### **Q5:**
> We added "estimation method" as a reporting field in Sec. 4.1. On the burden concern , we want to clarify that the DIA schema handles this through the data_quality field.
>
> ### **Q6:**
> Added this argument to Sec. 3: tracking environmental costs is valuable as a matter of scientific best practice, independent of whether reporting alone reduces impact.
>
> ### **Q7:**
> Addressed this in W2. The worked example in Appendix D.2 shows 3 formats: YAML metadata for HF model cards (machine-readable, parseable by dashboards), a one-line entry for paper reproducibility checklists, and a data_quality field flagging measured vs. estimated values.
>
> We thank the reviewer again ,these suggestions have meaningfully improved the paper.

---

> > ### Author Rebuttal · Reviewer_5ETx · 2026-04-02
> >
> > Thank you for your answers and your committment to further improve the paper. I look forward to it. I have also raised my score accordingly.

---

### Official Review · Reviewer_7nMA · 2026-03-13

**Significance:** 3
**Argument Clarity:** 3
**Rating:** 5
**Confidence:** 3

**Questions:**

The DIA framework is more diagnostic; it doesn't seem to come with a specific lever that would inevitably lead to reduced environmental footprints. Do you agree? Do you have any ideas how this could be achieved?

**Alternative Views Section:**

Yes

**Compliance With Llm Reviewing Policy A Conservative:**

Affirmed.

**Discussion Potential:**

4

**Final Justification:**

Overall, I'm still not very convinced by the provided evidence that the rebound effect is true in AI, as the references only describe how usage/model derivatives grew (or may grow in the future) but don't really connect them to efficiency improvements or even show that it's caused by improved efficiency. Regardless of whether the AI rebound effect is true, the paper is valuable, and I'm happy with the other proposed revisions, so I increased my score from 4 to 5.

**Paper Summary:**

The paper emphasizes that while the environmental impact of (pre-) training large AI models is often the center of focus, the aggregate footprint due to extensive use and fine-tuning of these models, especially open-source ones, tends to be underreported. The authors argue that this is problematic, since more efficient base models may trigger a 'rebound effect' that actually increases the overall environmental footprint. The stated position is, thus, that it's important to track the *ecosystem-level* footprint of open-source models. The proposed Data and Impact Accounting (DIA) framework described how this could be implemented in practice, including by establishing low-friction measurement tools and aggregation of ecosystem-level impacts via public dashboards.

**Position:**

Yes

**Position In Title:**

Yes

**Related Work:**

4

**Strengths And Weaknesses:**

Strengths:
- Position is well supported and timely. The aggregate, hard to measure, environmental footprint of open-source model derivatives is an important problem and the paper's shifting of the focus from model-specific efficiency to ecosystem-level cumulative impacts is welcome and could inspire discussion in the broader community.
- Well written, clear argumentation.
- Alternative views are comprehensively and convincingly discussed.

Weaknesses:
- The paper's core premise is the 'rebound effect', i.e. how model efficiency boosts drive adoption and increased environmental footprints. The paper provides some good intuition on why this may be true (e.g. similar things have been observed for cloud computing), however having some concrete evidence (even if somewhat anecdotical) would strengthen the paper, since this premise is key to the paper's position.
- Relatedly, when stating the estimated increase in data center energy consumption, the paper's conclusion (*"This pattern is consistent
with a rebound effect, where efficiency improvements reduce the cost of AI inference, which increases demand and,
in turn, drives more supply"*) seems a bit one-sided to me. The increase in usage may also or even mostly be due to model performance boosts and other non-efficiency related benefits?
- Not clear how figure 1 shows *"cumulative, often invisible, impact of derivative models and the rebound effect"* or how *"AI’s footprint extends beyond base training to include downstream derivatives"*.
- Table 2: Please describe in the appendix how exactly the estimated quantities were calculated/extrapolated.

**Support:**

2

---

> ### Author Rebuttal · Authors · 2026-03-27
>
> We thank Reviewer 7nMA for the constructive feedback and for recognizing the discussion potential and related work coverage. We address each one of those questions and suggestions below:
>
> ### **Regarding W1 - Rebound Effect Needs Empirical Evidence:**
>
> We thank the reviewer for the useful suggestion. We have added concrete empirical anchors:
> - Google reported a 48% increase in GHG emissions from 2019 - 2023 that attributes the rise to data centre energy driven by AI workloads, despite simultaneous PUE and per-query efficiency improvements (Google 2024 Environmental Report).
> - IEA data shows AI-specific servers growing at ~30% annually even as per-query energy has fallen to sub-watt-hour levels (IEA, 2025).
>
> - Sharma (2024) documents the Jevons Paradox in cloud computing: reduced per-unit costs led to aggregate demand growth that outpaced efficiency gains that were structurally analogous to AI derivatives.
>
> - Laufer et al. (2025) documents 146 derivatives for a single model family, illustrating how individually modest costs compound.
>
> (IEA 2025) https://iea.blob.core.windows.net/assets/dd7c2387-2f60-4b60-8c5f-6563b6aa1e4c/EnergyandAI.pdf
> Sharma (2024) https://arxiv.org/abs/2411.11540
> Laufer et al. (2025)  https://arxiv.org/abs/2508.06811
>
> We have added these evidence and connect them with LLM derivatives in the manuscript.
>
> ### **Regarding W2 - Energy Increase May Have Non-Efficiency Causes:**
>
> We agree that the growth in energy consumption should not be directly attributable to rebound effects alone. Model capability improvements, expanding user adoption, and new application domains (e.g., agentic AI, multimodal systems) are also major drivers. We have **revised Section 2.1 to reflect this multi-factor framing**.
>
> ### **Regarding W3 -  Figure 1 Needs Clarification:**
>
> We apologize for the confusion. The original caption overstated what Figure 1 directly illustrates. The figure shows base model footprints and regional water stress, but does not depict derivatives or the rebound effect. **We have revised the caption to accurately reflect the figure's content** and removed the unsupported claims.
>
> ### **Regarding W4 - Table 2 Calculation Methodology:**
>
> We appreciate this suggestion. We want to mention that the estimation methodology was already documented in Appendix A (Equations 1–3) and now we have enhanced on Appendix C - GPT-4 worked example and in the revision, **we have added a step-by-step worked example for DeepSeek-V3** ( newly added Appendix D.1 "Worked example”) **and a sensitivity analysis (Table A5 in Appendix D)** showing how estimates vary by 3-8× under different assumptions for utilization, PUE, and carbon intensity.
>
> For DeepSeek-V3, we reported 2.79M H800 GPU-hours × 350W (50% of 700W TDP) × PUE 1.1 yields ~1.07 GWh, giving ~548 tCO₂eq at CI=0.51 kgCO₂/kWh (consistent with Table 1). The sensitivity analysis shows estimates vary by 3 - 8× across assumptions, for example, GPT-4 ranges from 1,584 tCO₂eq (conservative: 60% util, PUE 1.1, CI 0.1) to 12,816 tCO₂eq (upper: full TDP, PUE 1.2, CI 0.445). DeepSeek-V3 ranges from 301 to 984 tCO₂eq. The largest uncertainty source is grid carbon intensity, followed by GPU utilization that highlights why standardized reporting via DIA matters.
>
> ### **Regarding Q - DIA as Diagnostic Tool:**
>
> We agree that DIA is diagnostic, not prescriptive, and we see this as a necessary foundation rather than a limitation. Reducing environmental impact requires first making it visible: without measurement, there is no meaningful basis for comparison or change. Once footprints are visible, practical changes follow naturally, for example, a researcher who sees that one training approach uses 50× more energy than another for similar results will pick the cheaper option.
>
> Model hubs that display energy cost next to accuracy let users make informed choices. Over time, this creates community pressure toward efficiency , this is much like how nutrition labels didn't force anyone to eat differently, but made it possible to compare and choose. If stronger measures like compute budgets are ever needed, they will depend on exactly the kind of measurement infrastructure DIA provides.
>
> We thank the reviewer again for the thoughtful feedback. These suggestions have strengthened the paper, and we hope the revisions address the concerns raised.

---

> > ### Author Rebuttal · Reviewer_7nMA · 2026-04-03
> >
> > I appreciate the author's response. About the first point, I'd like to invite the authors to be very precise when incorporating these matters into their revised paper. In particular:
> > 1) The IEA report *projects* AI-specific electricity consumption to grow by 30% annually. Even though this is their base case, it seems too bold to describe it with "shows", which suggests it is a fact.
> >
> > 2) The Laufer el al. (2025) paper says nothing about energy consumption, so I'd be more careful if using this as a reference for the rebound effect and emphasize that it's your interpretation of what the paper implies.
> >
> > Overall, I'm still not very convinced by the provided evidence that the rebound effect is true in AI, as the references only describe how usage/model derivatives grew (or may grow in the future) but don't really connect them to efficiency improvements or even show that it's caused by improved efficiency. Regardless of whether the AI rebound effect is true, the paper is valuable, and I'm happy with the other proposed revisions, so I increase my score.

---

### Decision · Program_Chairs · 2026-04-30

**Decision:**

Accept (spotlight)

**Comment:**

The paper addresses a timely and somewhat controversial topic of the environmental sustainability of closed vs open source models, addressing an important gap in the prevailing community discourse on this topic. The paper is well written, the evidence is compelling, and the proposed framework provides a clear and actionable path forward.